# Phase Imbalance Optimization in Interference Linear Displacement Sensor with Surface Gratings

**DOI:** 10.3390/s20051453

**Published:** 2020-03-06

**Authors:** Sergey Odinokov, Maria Shishova, Michael Kovalev, Alexander Zherdev, Dmitrii Lushnikov

**Affiliations:** Bauman Moscow State Technical University, 2nd Baumanskaya st. 5/1, 105005 Moscow, Russia; mshishova@bmstu.ru (M.S.); m.s.kovalev@gmail.com (M.K.); zherdev@bmstu.ru (A.Z.); dmlu41@yandex.ru (D.L.)

**Keywords:** diffraction grating, holographic grating, linear displacement sensor, phase imbalance, holographic materials, interference system

## Abstract

In interferential linear displacement sensors, accurate information about the position of the reading head is calculated out of a pair of quadrature (sine and cosine) signals. In double grating interference schemes, diffraction gratings combine the function of beam splitters and phase retardation devices. Specifically, the reference diffraction grating is located in the reading head and regulates the phase shifts in diffraction orders. Measurement diffraction grating moves along with the object and provides correspondence to the displacement coordinate. To stabilize the phase imbalance in the output quadrature signals of the sensor, we propose to calculate and optimize the parameters of these gratings, based not only on the energetic analysis, but along with phase relationships in diffraction orders. The optimization method is based on rigorous coupled-wave analysis simulation of the phase shifts of light in diffraction orders in the optical system. The phase properties of the reference diffraction grating in the interferential sensor are studied. It is confirmed that the possibility of quadrature modulation depends on parameters of static reference scale. The implemented optimization criteria are formulated in accordance with the signal generation process in the optical branch. Phase imbalance and amplification coefficients are derived from Heydemann elliptic correction and expressed through the diffraction efficiencies and phase retardations of the reference scale. The phase imbalance of the obtained quadrature signals is estimated in ellipticity correction terms depending on the uncertainties of influencing parameters.

## 1. Introduction

Linear displacement sensors with fixed reading head determine the current coordinate and displacement direction of an object connected with the measurement scale [1,2,3]. Optical interferential sensors show the highest resolution and accuracy for linear displacement measurements. There are many optical schemes that form two sinusoidal signals shifted in phase by π/2 and implement a quadrature detector with the help of optical interference. The arctangent of the ratio of these signals amplitudes is a linear function within one period, wherefrom the current location of the measurement scale is calculated. Subnanometer resolution levels can be achieved by subsequent interpolation of this function. In order for the interpolation to achieve high resolution, it is necessary to generate the pair of quadrature signals as perfectly as possible. The main problems of quadrature modulation are associated with signal distortions such as the inequality of amplitudes, shift of zero levels (offset), and the deviation of the phase shift from the π/2, i.e., phase imbalance. Last error is measured in electrical degrees. The amplitude imbalance can be eliminated by application of differential scheme. In this paper, we investigate phase retardations occurring in optical branch and achievable phase balance levels. We present the rationale of correct quadrature modulation in linear displacement sensor with two holographic gratings. In this regard, it is necessary to control the phase of the optical components not excluding the gain monitoring of the detectors.

Although quadrature modulation is widely used in grating interferometers, only a few publications discuss the corresponding problems for creating quadrature phase shift [3,4,5]. At present, the problem of eliminating the phase imbalance of quadrature signals in the optical branch remains unsolved. Correction of this type of distortion is carried out electronically in the presence of an acceptable value, but can be minimized optically, which simplifies signal processing in real time.

The proposed method may be useful in other areas of interferometric measurements, specifically for biological applications [6,7] and coherence tomography [8,9]. In shearing interferometry, optimization of the reference element parameters can help to perform quantitative linear phase retardance [10,11,12]. The presented analysis of laser radiation conversion in the interferometer may improve phase unwrapping of interferometric fringes [13] or multistep phase-shift extraction [14,15]. The method can be applied to uncertainty precalculation before experimental implementation of digital methods as digital holographic interferometry [16,17] or digital speckle interferometry [11]. Thereby, the studied method is not limited to displacement measurements, and can be applied to optimize phase relationships in any interferometric schemes with diffractive or holographic elements.

The article is organized as follows. Section 2 presents an optical concept and quadrature signals generation in the optical branch. Section 3 gives the optimization condition for the formation of quadrature signals. To simulate the conversions of optical radiation by diffraction gratings, a rigorous coupled waves analysis (RCWA) was used. Phase imbalance and amplification coefficients are deduced using Heydemann elliptic correction [18,19]. Section 4 describes the application of the proposed method to identify an instrumental uncertainty of double grating linear displacement sensor by estimating phase imbalance in the presence of various uncertainty contributions in the optical scheme.

## 2. Materials and Methods

Homodyne [20] and heterodyne implementations [21] are widely used among the interferential schemes for linear displacement sensors. In this research, we analyze the homodyne interferometer with two diffraction gratings [22] and consider the problem of minimizing phase imbalance and amplification correction of quadrature signals. The implementation of double grating scheme allows the beams to remain parallel inside the system due to the consistency of the diffraction directions. Therefore, the alignment of this optical systems is simple if compared to ones where the reference grating is replaced by a group of refractive optical components [23]. These features leave phase imbalance to be associated with instrumental characteristics in the linear displacement sensor.

### 2.1. Optical Concept

Figure 1 illustrates operation diagram of linear displacement sensor in conjunction with a nanopositioning moving stage including signal generation. The measurement scale is mounted on a movable platform with a non-contact linear drive and the reading head is mounted motionless perpendicularly to the direction of movement. The reading head must be static so as not to transmit any vibrations caused by wire connections to the electronic unit interface.

To determine the linear displacement, the measurement scale moves relative to the reading head of the sensor, thereby forming changes in the signals (Figure 1b) obtained by the interference of laser beams. When the reading head detects these changes, it generates four electrical sinusoidal signals, which must be phase shifted to 90° from each other. Movement is carried out in the direction of the *x* axis. Figure 1a–c shows this process conventionally. Subsequent subtraction of their pairwise antiphase leads to the amplitude doubling and elimination of the mismatch of the offset levels. This allows obtaining an updated pair of quadrature signals. These analog sine and cosine signals are then converted and interpolated with a high factor, e.g., 14 bits or 2^14^ = 16,384. However, displacement determination along with interpolation can be accurate if only the original signals are generated without errors. The interpolation error is defined as an error associated with deviations of the working conditions from the quadrature signals and directly depends on the phase imbalance. The interpolation error is cyclic and occurs with each period of the signal, but it does not accumulate or depend on the period value of the diffraction scale. At the last step, before interpolation and determination of displacement, correction of residual ellipticity is carried out, if it is not eliminated in the optical branch, as well as for the correction of random phase deviations in real time. The signal errors depending on the positioning of elements in the optical system in accordance with Abbe errors [24,25] is not the subject of the current research, although it is very important in the practical realization. The Abbe errors need to be aligned in the scheme to avoid interference break in the plane of the photodetectors, where it is necessary to provide an interference strip of infinite wide. Therefore, the durable angle of relative inclination between the scales should not exceed 0.5°. To minimize Abbé errors and the most influential pitching error measurement bench should be provided with high-precision linear guides. In addition, mounting surfaces should be sufficiently machined causing no additional inaccuracies in the system. General recommendations on minimizing the angular errors include moving the heavier load as close as possible to the center of the measurement system.

### 2.2. Quadrature Signals Generation in the Optical Branch

Figure 2a shows the ray path in the optical scheme of the sensor and the formation of quadrature signals by interference with linear displacement of the measuring scale. Only those beams that forms quadrature signals are shown. However there are other diffraction directions provided by diffraction grating Equation (1) and not used in the optical system in which diffraction efficiency should be minimized. The beam splitting on reference and measurement scales is carried out in accordance with the grating equation
(1)$$d(\sin\theta_{inc}+\sin\theta_{m})=m\lambda$$
where *d* is the period of the diffraction grating, θ*_inc_* is the incident angle, θ*_m_* is the diffraction angle of the *m*th order, and *λ* is the laser wavelength. The ratio of the period of the structure and the wavelength determine the scheme geometry of the circuit.

The laser radiation is collimated, and after diffraction on the reference grating, it is divided into three beams. Then, each of these beams is diffracted on the measurement scale in reflection, and four operation beams are formed. They again fall on the reference grating and, after third diffraction, interfere two by two. Four photodiodes arranged according to the propagation directions of the interfering beams receive signals that have sinusoidal dependence on the displacement. Multiple reflections are present on the glass surfaces according to Fresnel equations. The distance between the diffraction scales is enough to avoid it intervening in the operation directions of light distribution. In particular, distance between the scales is calculated based on the photodetector size and laser beam aperture. To avoid spurious diffraction orders arriving to the photodetectors, the distance between the scales few times exceeds the laser beam aperture. If the optical system is aligned, an interference band of infinite width is observed at every photodetector, as shown in Figure 2b. The basic principle of displacement measuring here is the phase coding of the optical signal during diffraction on a measurement scale. The diffracted beam acquires a certain phase shift Ω, depending on the displacement *x* of the measurement scale with a period *d*: Ω = ±2π*x*/*d*. That phase shift makes it possible to introduce motion information into optical signals. To apply the principle of quadrature modulation, it is necessary to arrange the 90° phase difference between the electrical signals received from photodetectors, as shown in Figure 2c–e. Quadrature modulation is generated using static components.

## 3. Results

Tracing the path of each of the interfering beams considers the process of amplitude dividing and phase transformation in detail. At each diffraction, in addition to the amplitude conversion of light in accordance with diffraction efficiency, the wavefront undergoes a certain phase shift [26,27]. Table 1 presents the structure of the successive transformations in each of the beams and the parameters of each diffraction are introduced: diffraction efficiencies η and phase shifts Φ.

To consider the interference of a pair of signals, it is convenient to introduce the complex wave amplitude in an exponential form. The initial complex wave amplitude after the collimator is expressed as *E_i_*_nc_ = *E*_0_ × exp(*I* × Φ_0_). Then, after the first diffraction three operating waves are formed on the reference grating with amplitudes *E*_11_ = η_11_ × *E*_0_ × exp(*I* × [Φ_0_ + Φ_11_]), *E*_12_ = η_12_ × *E*_0_·exp(*i* × [Φ_0_ + Φ_12_]) and E_13_ = η_13_
*E*_0_ × exp(*I* × [Φ_0_ + Φ_13_]). During the second diffraction on the measurement scale, the dependence on the displacement and the phase shift during diffraction are introduced into the signal; forming four signals with complex amplitudes *E*_21_(*x*) = η_21_ × η_11_ × *E*_0_ × exp(*i* × [Φ_0_ + Φ_11_ + Φ_21_ − Ω(*x*)]), *E*_22_(*x*) = η_22_ × η_12_ × *E*_0_ ×exp(*I* × [Φ_0_ + Φ_12_ + Φ_22_ + Ω(*x*)]), *E*_23_(*x*) = η_23_ × η_12_ × *E*_0_ × exp(*i* × [Φ_0_ + Φ_12_ + Φ_23_ − Ω(*x*)]), *E*_24_(*x*) = η_24_ × η_13_ E_0_ × exp(*I* × [Φ_0_ + Φ_13_ + Φ_24_ + Ω(*x*)]). These signals have become dependent on displacement value. After the third diffraction, each of these waves is subdivided into two, forming four pairs of signals *E*_31_(*x*) and *E*_34_(*x*), *E*_32_(*x*) and *E*_33_(*x*), *E*_35_(*x*) and *E*_38_(*x*), and *E*_36_(*x*) and *E*_37_(*x*), which are superimposed due to the interference.

The complex amplitude of the wave arriving at the first detector PD1 is
(2)$$\begin{array}{cc} E_{\mathrm{PD}1}\left(x\right) & =\eta_{32}\eta_{21}\eta_{11}E_{0}\exp\left(i\left[\Phi_{0}+\Phi_{11}+\Phi_{21}-\Omega (x)+\Phi_{32}\right]\right)+\\ & \eta_{32}\eta_{22}\eta_{12}E_{0}\exp\left(i\left[\Phi_{0}+\Phi_{12}+\Phi_{22}+\Omega (x)+\Phi_{33}\right]\right) \end{array}$$
Then, an intensity signal that is also equivalent to a voltage signal is
(3)$$I_{\mathrm{PD}1}\left(x\right)=E_{\mathrm{PD}1}\times E_{\mathrm{PD}1}{}^{*}=\frac{1+V_{\mathrm{PD}1}\cos(\Psi_{{}_{\mathrm{PD}1}})}{A_{\mathrm{PD}1}{}^{2}+B_{\mathrm{PD}1}{}^{2}},$$
where A_PD1_ = η_32_ × η_21_ × η_11_ × E_0_, B_PD1_ = η_33_ × η_22_ × η_12_ × E_0_, V_PD1_ = 2 A_PD1_ × B_PD1_ /(A_PD1_^2^ + B_PD1_^2^) = 2η_32_ × η_21_ × η_11_ × η_33_ × η_22_ × η_12_/[(η_32_ η_21_ η_11_)^2^ + (η_33_ η_22_ η_12_)^2^], and Ψ_PD1_ = Φ_32_ + Φ_21_ + Φ_11_ − Φ_33_ − Φ_22_ − Φ_12_ − 2Ω(*x*).

Similarly, we determine the optical signals *I*_PD2_, *I*_PD3_, and *I*_PD4_ arriving to the remaining detectors A_PD2_ = η_31_ η_21_ η_11_ E_0_, B_PD2_ = η_34_ η_22_ η_12_ E_0_, Ψ_PD2_ = Φ_31_ + Φ_21_ + Φ_11_ − Φ_34_ − Φ_22_ − Φ_12_ − 2Ω(*x*); A_PD3_ = η_35_ η_23_ η_12_ E_0_, B_PD3_ = η_38_ η_24_ η_13_ E_0_, Ψ_PD3_ = Φ_38_ + Φ_24_ + Φ_13_ − Φ_35_ − Φ_23_ − Φ_12_ + 2Ω(*x*); A_PD4_ = η_36_ η_23_ η_12_ E_0_, B_PD4_ = η_37_ η_24_ η_13_ E_0_, Ψ_PD4_ = Φ_13_ + Φ_24_ + Φ_37_ − Φ_12_ − Φ_23_ − Φ_36_ + 2Ω(*x*). The received four signals *I*_PD1_–*I*_PD4_ are shown schematically in Figure 2c. Due to the symmetry condition of a sinusoidal or rectangular profile of the grating, the diffraction efficiencies and optical phase shifts will be equal under symmetric diffraction conditions
η_11_ = η_13_,η_22_ = η_23_, η_21_ = η_24_, η_31_ = η_38_, η_32_ = η_37_, η_33_ = η_36_, η_34_ = η_35_;Φ_11_ = Φ_13_, Φ_22_ = Φ_23_, Φ_21_ = Φ_24_, Φ_31_ = Φ_38_, Φ_32_ = Φ_37_, Φ_33_ = Φ_36_, Φ_34_ = Φ_35_.(4)

It should be noted that the interference contrast *V* is determined by six values of diffraction efficiency. The current phase is determined by six values of the phases introduced during diffractions, and the doubled phase, depending on the displacement. The presence of 2Ω(*x*) indicates that, in the studied optical scheme, the signals period is half the period of the measurement scale. Currently, scientists can achieve eight-fold optical division of the period of the measurement scale that provides a corresponding increase in resolution [28]. For any interferential implementation, the laser beam arriving at the reflective measurement scale, within the entire contact surface, must cover the area of the desired period to be sure of the correct phase shifting within the displacement period; therefore, the diffraction grating should be represented as a qualitatively uniform structure of a constant period without frames. It is possible to make a measurement scale with large frames with help of a two-probe scheme [29], when the reading head is designed in such a way to generate and receive two sets of beams separated in space. Each signal set is responsible for own control zone on the measurement scale. Then, when passing the frame border, one of the beams always falls into the region with a uniform period within one frame.

To ensure quadrature modulation, it is necessary for the phase difference between I_PD1_ and I_PD2_ to be 90°
ΔΨ_1-2_ = Ψ_PD1_ − Ψ_PD2_ = Φ_32_ + Φ_21_ + Φ_11_ − Φ_33_ −Φ_22_ −Φ_12_ − 2Ω(*x*) − − [Φ_31_ + Φ_21_ + Φ_11_ − Φ_34_ − Φ_22_ −Φ_12_ − 2Ω(*x*)] = Φ_32_ + Φ_34_ − Φ_33_ − Φ_31_ = 90°(5)

It turned out that the final phase difference, and therefore quadrature modulation, is determined by the third diffraction inside of the double grating optical system. By generating and controlling the phase shifts introduced during this third diffraction, we set the requirements for the reference grating. In this regard, we are able to control the quadrature modulation in the optical branch without introducing additional optical phase components. The combined energy and phase calculation of the diffracted waves helps to create the quadrature modulation. The two remaining signals from the photodiodes PD3 and PD4 must be out of phase in order to effectively reset the offset: ΔΨ_1-4_ = Ψ_PD1_–Ψ_PD4_ = Φ_11_ + Φ_21_ + Φ_32_ − Φ_12_ − Φ_22_ − Φ_33_ − 2Ω(*x*) − [Φ_13_ + Φ_24_ + Φ_37_ − Φ_12_ − Φ_23_ − Φ_36_+2Ω(*x*)] = −4Ω(*x*). Therefore, Ψ_PD4_ = Ψ_PD1_ + 4Ω(*x*). ΔΨ_2-3_ = Ψ_PD2_ − Ψ_PD3_ = Φ_11_ + Φ_21_ + Φ_31_ − Φ_12_ − Φ_22_ − Φ_34_ − 2Ω(*x*) − [Φ_13_ + Φ_24_ + Φ_38_ − Φ_12_ − Φ_23_ − Φ_35_ + 2Ω(*x*)] = −4Ω(*x*). Therefore, Ψ_PD3_ = Ψ_PD2_ + 4Ω(*x*). The antiphase signals are determined by various signs in front of the phase term containing the cosine function.

Distorted signals after the subtraction are
*U_cos_*∝ *I_cos_* = I_PD1_ − I_PD4_ = *p* + *R_cos_* cos(2Ω(*x*))*U_sin_*∝*I_sin_* = I_PD2_ − I_PD3_ = *q* + *R_sin_* sin(2Ω(*x*) − α)(6)
where *p* = (η_32_ η_21_ η_11_ E_0_)^2^ + (η_33_ η_22_ η_12_ E_0_)^2^ − (η_36_ η_23_ η_12_ E_0_)^2^ − (η_37_ η_24_ η_13_ E_0_)^2^→0; *R_cos_* = 2E_0_^2^(η_32_ η_21_ η_11_ η_33_ η_22_ η_12_ + η_36_ η_23_ η_12_ η_37_ η_24_ η_13_) → 4η_32_ η_21_ η_11_ η_33_ η_22_ η_12_ E_0_^2^; *q* = (η_31_ η_21_ η_11_ E_0_)^2^ + (η_34_ η_22_ η_12_ E_0_)^2^ − (η_35_ η_23_ η_12_ E_0_)^2^ − (η_38_ η_24_ η_13_)^2^→0; *R_sin_* = 2E_0_^2^(η_31_ η_21_ η_11_ η_34_ η_22_ η_12_ + η_35_ η_23_ η_12_ η_38_ η_24_ η_13_) → 4 η_31_ η_21_ η_11_ η_34_ η_22_ η_12_ E_0_^2^; and α = |Φ_32_ + Φ_34_ − Φ_33_ − Φ_31_ − 90°|→0°.

*R_cos_* and *R_sin_* are the amplitudes of the signals, which relation defines the needed amplification in the channels; *p* and *q* are the offset levels of the sine and cosine channels, respectively; and α is the quadrature error or phase imbalance. When the condition in Equation (5) is fulfilled, the signal amplitude doubles, and the offset level is zeroed (*p* →0*, q* →0). The advantage of this procedure, in addition to zeroing and equalizing the basic signal levels, is that it also consists in an increase in the signal-to-noise ratio. The period of the signals decreases by half relative to the period of the measurement scale; *p* and *q* are different if the diffraction efficiencies are different for different diffraction conditions. α expresses the residual ellipticity (imperfect quadrature alignment) that is zeroed out in the practical implementation of the condition in Equation (5)—successful adjustment of the phase imbalance due to proper manufacturing of the static reference scale. In electronic processing, we no longer work with the optical signals *I_cos_* and *I_sin_*, but with the proportional voltage signals *U_cos_* and *U_sin_*, which are schematically shown in Figure 2d.

### 3.1. Elliptical Correction Phase Analysis

When the signals are presented in the form of Equation (6), we can apply the Heydemann correction [19] of quadrature measurement errors in interferometers. For the interferential scheme in Figure 2, the perfect quadrature detection would be *U_cos_^perf^* = *R* cos(Ψ), *U_sin_^perf^* = *R* sin(Ψ), where = 2Ω(*x*) **=** 4π *x*/*d*, *R* = *R_cos_*
*= G R_sin_* is the radius of the Lissajous figure in the form of a circle, and *G = R_cos_*/*R_sin_*
*is the—*the amplification coefficient. Measurements of the instantaneous coordinate values based on instantaneous signals *U_cos_^perf^* и *U_cos_^perf^* are possible by the interpolation between the calculated points of zero intersections *x* in the Lissajous figure. These intersections correspond to a quarter of the period of the quadrature signal. Fractional distance values in phase representation Ψ are obtained from the arctangent function of the ratio of signal amplitudes. The main difficulty in digital adjustment of phase imbalance is to determine the coefficients *r*, *p*, *q*, and α from the experimental signal (Equation (6)), assuming that these are the only significant demodulation errors. To provide it, a sufficiently large distance *x* is covered, accumulating experimental data to determine these four components. This operation is carried out by selecting the least square method.

In a real system, the end of the vector (*U_cos_*, *U_sin_*) is not located on a circle, but on a distorted ellipse, which leads to significant errors in determining the fractional part Ψ of a quarter of the period. The circle (*U_cos_^perf^*)^2^ + (*U_cos_^perf^*)^2^ = *R*^2^ turns into an ellipse (*U_cos_*)^2^ + (*U_sin_*)^2^ = *R*^2^. The trajectory of the end of the vector (*U_cos_*, *U_sin_*) representing a distorted ellipse can be described according to
(7)$$\begin{array}{l} U_{\cos}=U_{\cos}^{perf};\\ U_{sin}=\frac{1}{G}(U_{sin}^{perf}\cos\alpha -U_{\cos}^{perf}\sin\alpha \end{array}$$
Then, the signals *U_cos_* and *U_sin_* (Figure 2d) can be corrected and reduced to the form *U’_cos_* and *U’_sin_* (Figure 2e), respectively
(8)$$\begin{array}{l} {U^{\prime }}_{\cos}=U_{\cos},\\ {U^{\prime }}_{\sin}=\frac{1}{\cos\alpha }(GU_{sin}+U_{\cos}\sin\alpha ) \end{array}$$
where G = η_32_ η_33_/η_31_ η_34_→1; α = |Φ_32_ + Φ_34_ − Φ_33_ − Φ_31_ − 90°|→0°.

To avoid problems caused by the multi-valued nature of this arctangent function, the current phase Ψ and displacement value ∆*x* is led to the following form
(9)$$\begin{array}{l} \Psi =\frac{1}{2}\left|\frac{U^{\prime }{}_{cos}}{\left|U^{\prime }{}_{cos}\right|}-\frac{U^{\prime }{}_{sin}}{\left|U^{\prime }{}_{sin}\right|}\right|+2\frac{U^{\prime }{}_{cos}}{\left|U^{\prime }{}_{cos}\right|}\cdot \frac{\arcsin U^{\prime }{}_{sin}}{\pi },\\ \Delta x=\left[{\left(x-\Psi \right)}_{1}-{\left(x+\Psi \right)}_{0}\right]\frac{d_{scale}}{8}, \end{array}$$
where indices 1 and 0 correspond to samples of zero intersections of *x* before and after the distance has changed. Encoder resolution and accuracy depend directly on how accurately *x* and Ψ are determined.

To formulate the requirements for the reference diffraction scale and determine whether it is possible to correct the ellipticity in the optical branch, it is necessary to simulate the diffraction problem. The optimization of the parameters of the reference scale is carried out in accordance with the condition
G = η_32_ η_33_/η_31_ η_34_→1; α = |Φ_32_ + Φ_34_ − Φ_33_ − Φ_31_ − 90°|→ 0°(10)

These conditions can be implemented when choosing the parameters of the reference scale—grating period and groove depth. At the same time, during the simulation and analysis, the polarization, wavelength, and shape of the grating relief are varied. The numerical solution of the diffraction problem and the corresponding calculation of the parameters of the reference scale η_31_, η_32_, η_33_, η_34_, Φ_31_, Φ_32_, Φ_33_, and Φ_34_ were carried out using MATLAB environment (ETMC Exponenta Ltd., Moscow, Russian Federation). To simulate the diffraction in sinusoidal grating, the RCWA method with the built-in method of curvilinear coordinates was used [30,31].

### 3.2. Phase Imbalance and Amplification in Quadrature Signals

Simulation for a sinusoidal diffraction scale is considered. The modulation zone of the diffraction grating is made of polymethyl methacrylate. The absolute value of the phase accumulated in the substrate of the diffraction grating does not affect the result of phase contribution calculation due to the symmetry of the optical scheme. The period of quadrature signals and, consequently, the period of the scales are selected from the division condition during interpolation with the powers of “2”, or based on the geometry of the optical scheme. Figure 3 shows the simulation results for the phase imbalance and amplification depending on the period and height of the reference diffraction scale. To analyze the phase imbalance and amplification in this work, simulation for a wide range of periods is carried out. Convenient propagation geometry corresponds to diffraction angles θ_1_ from 25° to 45° (Figure 2), which determines the periods for a laser wavelength using Equation (1). Different polarization conditions are considered; TE polarization is recommended. The groove depth in the analysis is limited by the maximum value of the period.

The value of the phase difference is restricted to the range from −180° to 180°; the contours at Figure 3 show the lines of the uniform phase imbalance in increments of 45°. It is important to note that the value of the phase imbalance, in contrast to the value of the amplification, keeps itself for different wavelengths while maintaining the diffraction angle. This suggests that it is possible to work with different sets of scales with different periods and corresponding coherent radiation sources in the same beam path geometry in the optical scheme.

For different laser wavelength the calculated reference scale parameters are: d_B_ = 0.751 μm, h_B_ = 0.67 μm for λ_B_ = 450 nm; d_G_ = 0.875 μm, h_G_ = 0.785 μm for λ_G_ = 0.52 μm; and d_R_ = 1.123 μm, h_R_ = 1.125 μm for λ_R_ = 0.66 μm. Furthermore, to analyze the contribution of the instrumental uncertainty in linear displacement encoder under study, the influence of manufacturing errors of reference scale and the stability of operating conditions are considered. It should be noted that, if these parameters do not match the calculated ones, it will be necessary to use an ellipticity correction in the interface electronics.

## 4. Discussion

The calculation of the diffraction gratings for double grating linear interference sensor includes: a light-energy calculation defining the conditions for the maximum interference contrast in photodetectors plane; calculation of phase relations according to the propagation of laser radiation inside the optical system; calculation of tolerance for the grating period and groove depth of the scales; and calculations of tolerances for refractive index changes and wavelength changes during operation. The input data of the optimization model are all the parameters necessary to solve the rigorous diffraction problem: parameters of the relief profile, spectral functions of the dielectric permittivity of the media used, and parameters of the incident radiation (wavelength, incidence vector, and polarization condition). Within the model framework, it is assumed that the optical system is aligned for observing the interference band of infinite width at photodetectors. It should be noted that consideration of conical diffraction allows analyzing the tolerances for angular displacements in the scheme, such as the rotation of the reference scale relative to the incident beam; accuracy of the laser adjustment is expressed in the illuminating angle deviation.

The proposed model allows calculating the uncertainty budget in the optical system related to the instrumental uncertainty. The contributions from the influencing parameters to the output signal are estimated by two key quadrature modulation characteristics: phase imbalance and amplification in the pair of channels. Usually, it is preferred to eliminate systematic uncertainty in possible ways. The proposed analysis allows evaluating the boundaries of permissible uncertainty values, knowing the manufacturing errors of the reference scale.

Figure 4 shows the dependencies of the phase imbalance and amplification, independent for each of the parameters affecting to the condition of uncertainty contributions. Four parameters were considered: period *d* and groove depth *h* of the reference scale, refractive index *n*, and wavelength λ. Uncertainties of the period Δ*d* and groove depth Δ*h* of the reference scale arise due to manufacturing inaccuracies. The wavelength uncertainty Δλ is associated with the heating of the radiation source and the corresponding change in the generation wavelength. The uncertainty of the refractive index Δ*n* is associated with the heating of the read head during the work. Functions α(*d*), α(*h*), α(λ), and α(*n*) allow us to estimate the contribution of the uncertainty to the phase imbalance, while the functions *G*(*d*), *G*(*h*), *G*(λ), and G(*n*) the contribution to the amplification.

In the presence of 2% uncertainties of each of the considered parameters: Δα (Δ*d*) = ± 2.5738°; Δ*G* (Δ*d*) = ± 0.0353 for Δ*d* = ± 0.01*d*; Δα (Δ*h*) = ± 0.2581°; ΔG (Δ*h*) = ± 0.0315 for Δ*h* = ± 0.01*h*; Δα (Δλ) = ± 2.4489°; Δ*G*(Δλ) = ± 0.038 for Δλ = ± 0.01λ; Δα(Δ*n*) = ± 0.0597°; and Δ*G*(Δ*n*) = ± 0.0011 for Δ*n* = ± 0.01*n*. In accordance with the above analysis, an assessment of the instrumental uncertainty is performed. Period and wavelength errors make the greatest contribution. The phase imbalance value is most sensitive to the period of the reference scale, whereas the amplification value is most sensitive to wavelength. In the considered double grating linear displacement sensor, exactly the groove parameters of the reference scale determine the ability to create the quadrature modulation. Operating conditions (temperature, humidity, and pressure) have a significant impact on the interferometer and must be normalized. Fluctuations of these parameters influence the system parameters as wavelength and refractive index of the scales, which was analyzed (Figure 4). This problem is also solved by manufacturing components from materials with a zero linear expansion coefficient (Sitall and Zerodur). The generalized uncertainties Δα of the phase imbalance function α(*d, h*, λ*, n*) and Δ*G* of the gain function in the channels *G*(*d, h*, λ*, n*) can be estimated in accordance with the equations
(11)$$\begin{array}{l} \Delta \alpha \left(\Delta d,\Delta h,\Delta \lambda ,\Delta n\right)=\sqrt{{\left(\frac{\partial \alpha }{\partial d}\Delta d\right)}^{2}+{\left(\frac{\partial \alpha }{\partial h}\Delta h\right)}^{2}+{\left(\frac{\partial \alpha }{\partial \lambda }\Delta \lambda \right)}^{2}+{\left(\frac{\partial \alpha }{\partial n}\Delta n\right)}^{2}},\\ \Delta G\left(\Delta d,\Delta h,\Delta \lambda ,\Delta n\right)=\sqrt{{\left(\frac{\partial G}{\partial d}\Delta d\right)}^{2}+{\left(\frac{\partial G}{\partial h}\Delta h\right)}^{2}+{\left(\frac{\partial G}{\partial \lambda }\Delta \lambda \right)}^{2}+{\left(\frac{\partial G}{\partial n}\Delta n\right)}^{2}}. \end{array}$$

The generalized uncertainties in the presence of 2% deviations of the considered parameters, as visualized in Figure 4, are Δα(Δ*d,* Δ*h*, Δλ*,* Δ*n*) = 3.5626° and Δ*G*(Δ*d,* Δ*h*, Δλ*,* Δ*n*) = 0.0475. The proposed method of analysis and optimization justifies the errors associated with the phase imbalance, allowing to predict the value for subsequent compensation. The correction of ellipticity becomes more effective as the optimization conditions are more accurately fulfilled in practice. The accuracy of the method under study is determined by the iteration step chosen for the calculation. In the current simulation (Figure 3 and Figure 4), this step was 1 nm. The chosen iterative step for each influential parameter should be several times smaller than the errors of the real diffraction scale. The limitations of the optimization method in practical implementation are related to manufacturing capabilities. Optical structures, manufactured according to calculation, provide good usability and performance for quadrature modulation. The resulting value of the phase imbalance provided by the fabricated reference grating with deviated parameters defines the practical limit of double-grating optical quadrature homodyne interferometry.

## 5. Conclusions

For reliable bi-directional motion detection, many interferometers generate two sinusoidal electrical outputs, which ideally should be equal in amplitude, have no relative offset, and have a phase difference of 90°. In practice, these criteria are not realized, and, therefore, it is necessary to apply corrections to quadrature phase signals to obtain nanometric resolution. This paper discusses a method to minimize these amendments. Deficiencies in the optical system of the sensor and, therefore, in quadrature signals are the main cause of the interpolation error. Even in high quality encoders, the interpolation error can range 1–2% of the signal period. Therefore, the interface electronics must include amplification, level adjustment, and phase balance of the signal to withstand interpolation errors. Application of the described method for analyzing the phase relations in the linear displacement sensor allows one to optimize the parameters of diffraction gratings in the optical scheme, as well as evaluate and eliminate the contribution of the instrumental uncertainty. Direct drive systems can operate with high gain control loop, which allows responding quickly to correct errors in position. However, with a high interpolation error, the frequency of the error increases, and the controller cannot cope with it. Then, the engine consumes more current, trying to react, which leads to noise and excessive heating of the engine. Understanding the instrumental error inside the sensor can simplify the requirements for interface electronics. The proposed analysis justifies errors associated with phase imbalance, thereby allowing us to predict the need for subsequent compensation. Elliptical correction becomes the more effective as the optimization conditions are more accurately fulfilled in the system. Using this approach, any of the schemes of homodyne or heterodyne displacement interferometers can be analyzed.

## Figures and Tables

**Figure 1 sensors-20-01453-f001:**
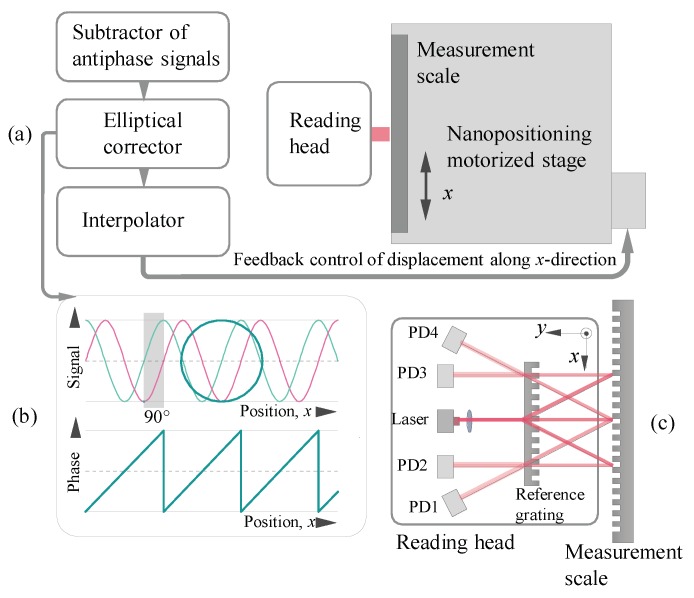
The scheme of the linear displacement sensor in conjunction with a moving stage (**a**); signal’s view after the elliptical correction (**b**); and optical system of double grating interferometer (**c**).

**Figure 2 sensors-20-01453-f002:**
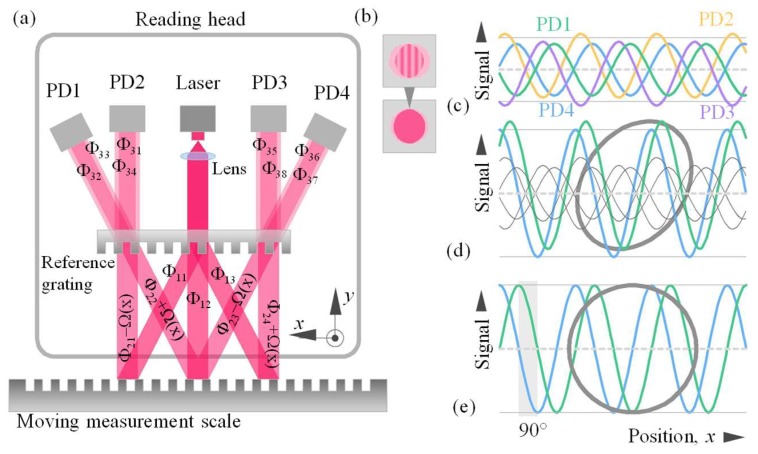
Signal transformation in the optical system: (**a**) the formation of signals due to ray path; (**b**) alignment for interference band of infinite width; (**c**) uncompensated signals; (**d**) signals after antiphase subtraction; and (**e**) signals after ellipticity correction.

**Figure 3 sensors-20-01453-f003:**
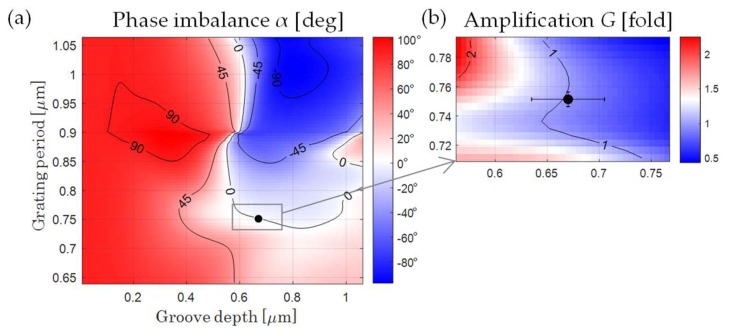
Phase imbalance α (**a**) and the amplification G between the channels (**b**) via period and groove depth of the reference grating at λ*_B_* = 450 nm.

**Figure 4 sensors-20-01453-f004:**
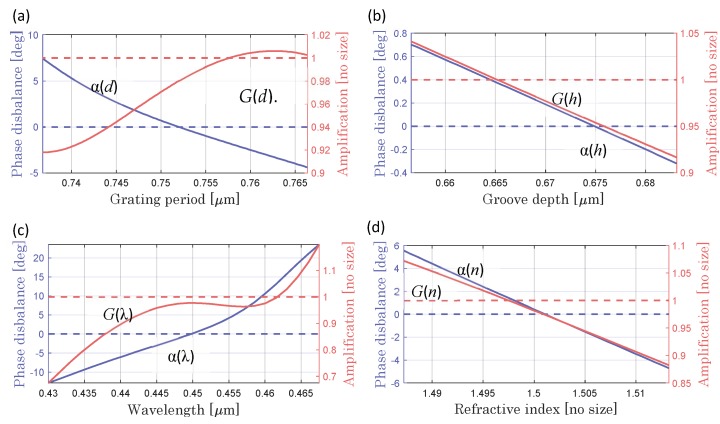
Phase imbalance α and amplification G of quadrature signals via period of: reference grating (**a**); groove depth of reference grating (**b**); laser wavelength (**c**); and refractive index (**d**). Dotted lines indicate optimal conditions.

**Table 1 sensors-20-01453-t001:** Sequence of dividing by amplitude and introducing phase shifts.

			1st diffraction on reference scale					
Order	−1		0				+1	
Diff. eff.	η_11_		η_12_				η_13_	
Phase shift	Φ_11_		Φ_12_				Φ_13_	
	2nd diffraction on measurement scale							
Order	−1		+1		−1		+1	
Diff. eff.	η_21_		η_22_		η_23_		η_24_	
Phase shift	Φ_21_ − Ω(x)		Φ_22_ + Ω(x)		Φ_23_ − Ω(x)		Φ_24_ + Ω(x)	
	3rd diffraction on reference scale							
Order	0	+1	0	−1	+1	0	−1	0
Diff. eff.	η_31_	η_32_	η_33_	η_34_	η_35_	η_36_	η_37_	η_38_
Phase shift	Φ_31_	Φ_32_	Φ_33_	Φ_34_	Φ_35_	Φ_36_	Φ_37_	Φ_38_
Detector	PD2	PD1	PD1	PD2	PD3	PD4	PD4	PD3

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
