# Peer review of "Phase Imbalance Optimization in Interference Linear Displacement Sensor with Surface Gratings"

_sensors, 2020, doi:10.3390/s20051453_

Round 1
Reviewer 1 Report
In this paper, authors study about phase imbalance of the gratings based interferometer. The sentences are well written and the figures as well. Basically, the paper has a high quality to give the acceptance. Here, I have some questions.
1. Although I am not sure about the distance between the two gratings (reference and measurement target), there must be multiple reflections. It seems that the authors did not consider the influences of the multiple reflections. Please add some discussion about that.
2. The last paragraph in section 3.1 was written in Russian(?). Please fix it.
3. During the stage displacement, there is a mechanical motion error such as pitching. The pitching error might induces the relative tilt between the two gratings. Is this tilt error able to ignore? How much is the durable angle of the pitching error?
Author Response
Dear reviewer,
Thank you for your time and comments that are very helpful to improve the article. Here are corrections according to your questions.
Point 1: Although I am not sure about the distance between the two gratings (reference and measurement target), there must be multiple reflections. It seems that the authors did not consider the influences of the multiple reflections. Please add some discussion about that.
Response 1: Multiple reflections are present on the glass surfaces according to Fresnel equations. However, the distance between the diffraction scales is enough to avoid it to intervene the operation directions of light distribution. In particular, distance between the scales is calculated based on the photodetector size and laser beam aperture. To avoid spurious diffraction orders arriving to the photodetectors the distance between the scales few times exceeds the laser beam aperture.
Point 2: The last paragraph in section 3.1 was written in Russian(?). Please fix it.
Response 2: The Russian paragraph duplicated the previous one. It is deleted. I’m very sorry for letting it happen.
Point 3: During the stage displacement, there is a mechanical motion error such as pitching. The pitching error might induces the relative tilt between the two gratings. Is this tilt error able to ignore? How much is the durable angle of the pitching error?
Response 3: The signal errors depending on the positioning of elements in the optical system in accordance with Abbe errors are very important in the practical realization [25, 26]. The Abbe errors needs to be aligned in the scheme to avoid interference break in the plane of the photodetectors, where it is necessary to provide an interference strip of infinite wide. Therefore, the durable angle of relative inclination between the scales should not exceed 0.5°. To minimize Abbé errors, and the most influential pitching error, measurement bench should be provided with high-precision linear guides. Also mounting surfaces should be sufficiently machined causing no additional inaccuracies in the system. General recommendations on minimizing the angular errors include moving the heavier load as close as possible to the center of the measurement system.
All the responses are also presented in the comments of review mode in the new version of the document.
Best regards, Authors
Reviewer 2 Report
Comments on the papers entitled, “Phase imbalance optimization in interference linear displacement sensor with surface gratings” by Maria Shishova et al.
General Comments:
The authors have proposed a method of the phase imbalance in the output quadrature signals of a sensor by calculating and optimizing the parameters of the gratings (in double grating interference schemes), based on the energetic analysis and also on phase relationships.
The proposed method may find its effectiveness in several interferometers and optical metrology. I found the method interesting. The manuscript is well documented and well written, however, there are some English and syntax errors.
Major Concerns:
There are some concerns listed below, that must be addressed and implemented before acceptance:
- There are many English and syntax errors, many to list here. So, the manuscript should be revised carefully.
- In the Abstract, provide the full form of RCWA.
- The Abstract should also carry finding information about the method.
- The authors should also provide the accuracy and limitations of the method.
- In the Introduction or later in the manuscript, the authors should provide a comprehensive overview of some state-of-arts and recently reported interferometers as the proposed method may be significant in many measurement interferometric methods/tools. Some of the recently published works have been provided below that should be cited in the manuscript:
- Kumar, Manoj, and Chandra Shakher. "Measurement of temperature and temperature distribution in gaseous flames by digital speckle pattern shearing interferometry using holographic optical element." Optics and Lasers in Engineering 73 (2015): 33-39.
- CGT Ruiz et al, “Cortical bone quality affectations and their strength impact analysis using holographic interferometry,” Biomedical Optics Express, Vol. 9, Issue 10, pp. 4818-4833 (2018)
- Kumar et al, ” Measurement of strain distribution in cortical bone around miniscrew implants used for orthodontic anchorage using digital speckle pattern interferometry,” Optical Engineering 55 (5), 054101.
- Wang et al., “Phase unwrapping of interferometric fringes based on a mutual information quality map and phase recovery strategy,” Optical Engineering, 57(11), 114108 (2018).
- Kumar, C Shakher, “Experimental characterization of the hygroscopic properties of wood during convective drying using digital holographic interferometry,” Applied Optics 55 (5), 960-968.
- Guerrero, A.L., Sainz, C., Perrin, H., Castell, R. and Calatroni, J., 1992. Refractive index distribution measurements by means of spectrally-resolved white-light interferometry. Optics & Laser Technology, 24(6), pp.333-339.
- Kumar et al., “Common-path multimodal three-dimensional fluorescence and phase imaging system,” Journal of Biomedical Optics 25 (3), 032010.
- Pedrini, G., Martínez-García, V., Weidmann, P., Wenzelburger, M., Killinger, A., Weber, U., Schmauder, S., Gadow, R. and Osten, W., 2016. Residual stress analysis of ceramic coating by laser ablation and digital holography. Experimental Mechanics, 56(5), pp.683-701.
- Dudescu, C., Naumann, J., Stockmann, M. and Nebel, S., 2006. Characterisation of thermal expansion coefficient of anisotropic materials by electronic speckle pattern interferometry. Strain, 42(3), pp.197-205.
- Mathematical expressions require a revision as some of the variables are not defined/written in the Russian language.
- On page number 7, the second paragraph must be revised as it is written in the Russian language.
- Figure 3, units of the color bar is missing.
- The Discussion Section should be described in detail. Figure 4 should also be discussed in detail.
Author Response
Dear reviewer,
Thank you for your time and comments that helped to significantly improve the article. Corrections according to your concerns are presented in the submitted dociment. All the responses are also in the comments of review mode in the new version of the manuscript.
Kind regards, Authors

Round 2
Reviewer 2 Report
In the revised manuscript, the authors have made satisfactory amendments in response to the raised concerns. The manuscript is suitable for publication, however, some substantial corrections are required such as Ref. 18 is repeated (with Ref. 11). I think, in place of Ref. 18, the following Ref. should be cited:
M. Kumar et al, “Measurement of strain distribution in cortical bone around miniscrew implants used for orthodontic anchorage using digital speckle pattern interferometry,” Optical Engineering 55 (5), 054101 (2016).